# Bio-Prospecting of Crude Leaf Extracts from Thirteen Plants of Brazilian Cerrado Biome on Human Glioma Cell Lines

**DOI:** 10.3390/molecules28031394

**Published:** 2023-02-01

**Authors:** Viviane A. O. Silva, Marcela N. Rosa, Izabela N. F. Gomes, Patrik da Silva Vital, Ana Laura V. Alves, Adriane F. Evangelista, Giovanna B. Longato, Adriana C. Carloni, Bruno G. Oliveira, Fernanda E. Pinto, Wanderson Romão, Allisson R. Rezende, Arali A. C. Araújo, Lohanna S. F. M. Oliveira, Alessandra A. M. Souza, Stephanie C. Oliveira, Rosy Iara Maciel A. Ribeiro, Rui M. Reis

**Affiliations:** 1Molecular Oncology Research Center, Barretos Cancer Hospital, São Paulo 14784-400, Brazil; 2Department of Pathology, School of Medicine of the Federal University of Bahia, Salvador 40170-110, Brazil; 3Gonçalo Moniz Institute, Oswaldo Cruz Foundation (IGM-FIOCRUZ/BA), Salvador 40296-710, Brazil; 4Research Laboratory in Cellular and Molecular Biology of Tumors and Bioactive Compounds, San Francisco University, Bragança Paulista, São Paulo 12916-900, Brazil; 5Petroleomic and Forensic Laboratory, Chemistry Department, Federal University of Espírito Santo, Vitória 29075-910, Brazil; 6Ituiutaba Unit, Department of Agrarian and Natural Sciences (DECAN), State University of Minas Gerais (UEMG), Divinopolis 38302-192, Brazil; 7Laboratory of Experimental Pathology, Federal University of São João del Rei—CCO/UFSJ, Divinópolis 35501-296, Brazil; 8Life and Health Sciences Research Institute (ICVS), School of Health Sciences, University of Minho, 4710-057 Braga, Portugal; 9ICVS/3B’s—PT Government Associate Laboratory, 4710-057 Braga, Portugal

**Keywords:** glioma, cytotoxic activity, cerrado biome, Melastomataceae, plant extracts

## Abstract

(1) Background: Malignant gliomas are aggressive tumors characterized by fast cellular growth and highly invasive properties. Despite all biological and clinical advances in therapy, the standard treatment remains essentially palliative. Therefore, searching for alternative therapies that minimize adverse symptoms and improve glioblastoma patients’ outcomes is imperative. Natural products represent an essential source in the discovery of such new drugs. Plants from the cerrado biome have been receiving increased attention due to the presence of secondary metabolites with significant therapeutic potential. (2) Aim: This study provides data on the cytotoxic potential of 13 leaf extracts obtained from plants of 5 families (Anacardiaceae, Annonaceae, Fabaceae, Melastomataceae e Siparunaceae) found in the Brazilian cerrado biome on a panel of 5 glioma cell lines and one normal astrocyte. (3) Methods: The effect of crude extracts on cell viability was evaluated by MTS assay. Mass spectrometry (ESI FT-ICR MS) was performed to identify the secondary metabolites classes presented in the crude extracts and partitions. (4) Results: Our results revealed the cytotoxic potential of Melastomataceae species *Miconia cuspidata, Miconia albicans,* and *Miconia chamissois*. Additionally, comparing the four partitions obtained from *M. chamissois* crude extract indicates that the chloroform partition had the greatest cytotoxic activity against the glioma cell lines. The partitions also showed a mean IC_50_ close to chemotherapy, temozolomide; nevertheless, lower toxicity against normal astrocytes. Analysis of secondary metabolites classes presented in these crude extracts and partitions indicates the presence of phenolic compounds. (5) Conclusions: These findings highlight *M. chamissois* chloroform partition as a promising component and may guide the search for the development of additional new anticancer therapies.

## 1. Introduction

Malignant gliomas are aggressive tumors characterized by fast cellular growth and highly invasive properties [1,2]. Among gliomas, glioblastoma is the most common and aggressive subtype [2]. The standard treatment option for glioblastoma consists of surgical resection, as wide as possible or partial, followed by a combined regimen of radiotherapy and adjuvant chemotherapy, mainly temozolomide (TMZ) [3,4,5,6]. Chemotherapy drugs approved for glioblastomas are limited. In addition to TMZ, there are carmustine, lomustine, and bevacizumab. However, clinically, TMZ is the standard drug for these tumors [7]. TMZ is a DNA alkylating chemotherapy drug that easily penetrates the blood-brain barrier [8,9]. It is important to emphasize due primarily to the over-expression of O6-methylguanine methyltransferase (MGMT) and/or lack of a DNA repair pathway in GBM cells, some patients do not respond to TMZ [10]. In addition, the severe side effects during the treatment turned the standard therapy, TMZ, still essentially palliative [1,11]. Therefore, research focusing on exploring novel therapeutic strategies for treatment that minimize adverse symptoms and improve glioblastoma patients’ outcomes is urgently needed [11,12].

Natural products are an important source for discovering new drugs [13]. Examples of medicinal plant-derived clinical antitumor agents include irinotecan, paclitaxel, etoposide, topotecan, camptothecin, and vincristine [13,14]. Many plant-derived antitumor drugs have been found through large-scale screening studies [15,16]. Brazil has been designated the world’s richest biodiversity, and Cerrado is considered the second largest biome, exhibiting a vast diversity of natural plants [17,18,19]. These plants bring particular properties that allow them to adapt to the hostile environment and the presence of herbivores and pathogens, reflecting the diversity of their secondary compounds [20,21].

Several studies have pointed out the biological effects of extracts from medicinal plants from the Cerrado, including anti-inflammatory, antioxidant, antiparasitic, and antitumor properties, as well as their use for gastrointestinal and respiratory disorders and wound healing [22,23,24,25,26,27,28,29].

De Melo et al. described a total of 84 plant species reported for use in the prevention or treatment of cancer in Brazil; 69.05% of these were mentioned as being used for the treatment of cancer in general and 30.95% for specific tumors. The most frequently mentioned plants were *Aloe vera*, *Euphorbia tirucalli*, and *Tabebuia impetiginosa*. At least one pharmacological study, in vivo or in vitro, was found for 35.71% of the species [30]. In addition, the authors showed that plants from the families Euphorbiaceae, Fabaceae, Apocynaceae, and Asteraceae are highlighted and are of great importance in ethnopharmacology studies aimed at experimental trials [30].

According to de Mesquita et al. (2009) [24], several compounds of Brazilian plant species from the cerrado have been studied with promising results. In their study, 28 out of 412 extracts tested demonstrated a substantial antiproliferative effect, with at least 85% inhibition of cell proliferation in one or more cell lines, using extracts obtained from different parts of Anacardiaceae, Annonaceae, Apocynaceae, Clusiaceae, Flacourtiaceae, Sapindaceae, Sapotaceae, Simaroubaceae, and Zingiberaceae. Furthermore, 50 of the 412 tested compounds are used in traditional medicine, and 21 families show an antineoplastic effect on tumor cell lines in vitro [24]. Despite these therapeutic activities, there were few reports about the effect of extracts isolated from cerrado plants [31], resulting in the cerrado biome as a potential candidate for bioprospecting efforts in cancer research [24,27,28,29].

Considering the potential of plants from Brazilian cerrado, in this study, we performed a phytochemical screening and provided data on the cytotoxic potential of leaf extracts from 13 plants of 5 families (Anacardiaceae, Annonaceae, Fabaceae, Melastomataceae, and Siparunaceae) found in Brazilian cerrado biome. Among these families described, there has been a research focus on Melastomataceae due to its biological potential therapeutic [32]. Therefore, the cytotoxic potential was evaluated on a panel of five glioma cell lines (including two adult glioblastomas, two pediatric gliomas, and one established primary glioblastoma cell culture) and one non-tumoral astrocyte. Our results indicate *Miconia chamissois* crude extracts and its chloroform partition as perspectives to develop new anticancer therapies.

## 2. Results

### 2.1. Cytotoxicity Profile Analysis of Crude Extracts

The in vitro cytotoxicity analysis of crude extracts was evaluated through the cell viability of five commercial cell lines: two adult glioblastomas (GAMG and U251-MG), two pediatric gliomas (SF188 and RES259), one normal human astrocyte (NHA) and one primary glioblastoma culture (HCB151). Initially, a screening was performed with 13 crude extracts of the Brazilian cerrado flora, which showed that the glioma cells presented a heterogeneous profile of cytotoxic response after 72 h of treatment (Table 1). Most extracts evaluated exhibited a dose-dependent response in glioma cells (Figure 1).

From the cell viability curves obtained of the effect of crude extracts concentrations on glioma cells, means of IC_50_ values ± standard deviation (SD) values for each cell line were calculated (Table 1). Some samples had their IC_50_ undetermined once it was below the lowest concentration evaluated (<1.5 µg/mL) or above the highest concentration evaluated (>200 µg/mL). Among the 13 extracts, crude extracts numbers 17, 18, and 19 from Melastomataceae *(M. cuspidata, M. albicans,* and *M. chamissois*, respectively) presented the highest cytotoxicity and a dose-dependent effect, with IC_50_ values close to the NCI criteria, which recommends IC_50_ values <30 μg/mL for compounds to be evaluated as promising and used for future purification tests [33]. The lowest in vitro inhibitory activity was observed for the *Bauhinia ungulata* (16-I), *Astronium fraxinifolium* (2), *Siparuna guianensis* (8), *Bauhinia variegata candida* (15-I) and *Xylopia aromatica* (3) crude extracts.

From the IC_50_ values obtained, it was also possible to evaluate the SI (Selectivity Index) for both crude extracts and the chemotherapy TMZ of glioma cells in relation to the normal astrocyte (NHA) (Table 2). Only crude extract 8 (*S. guianensis*) presented an expressive SI of 5.4 for the GAMG cell line (Table 2). It is important to emphasize that even the chemotherapy, TMZ, did not show selectivity for the tumor cell lines.

### 2.2. Cytotoxicity Profile Analysis of Partitions from Miconia Chamissois Crude Extracts

Among the 13 extracts, crude extracts 17, 18, and 19 from Melastomataceae were of the highest cytotoxicity; few studies published on the biological activity and chemical composition of *M. chamissois* have supported its therapeutic importance. Therefore, in order to identify the cytotoxic potential from secondary compounds of crude extract 19 (*M. chamissois)* in glioma cells, it was partitioned into HDP (hydroalcoholic partition), HP (hexane partition), CP (chloroform partition), and AEP (ethyl acetate partition). The result revealed that most partitions showed a higher cytotoxic response to glioma cells when compared to crude extract (Table 3). Among the partitions, CP was more cytotoxic than partitions HP and AEP after 72 h of treatment. However, for the HDP partition, it was not possible to determine the IC_50_ values for all cell lines evaluated, suggesting that they are greater than 200 µg/mL (Table 3).

Moreover, the results obtained from SI of partitions showed that AEP was 12.1 more selective for GAMG cells than normal astrocytes (NHA) (Table 4).

Finally, in order to identify the cell viability kinetics, glioma cells were exposed to *M. chamissois* crude extract and its partitions HP and CP at 6, 12, 24, and 48 h. The results showed that most glioma cells had a dose-and-time-dependent cytotoxic response to crude extracts and partitions (Table 5). Some cells had their IC_50_ undetermined once it was more resistant (>200 µg/mL or >300 µg/mL) than the highest concentration. Depending on the cell line, the greatest cytotoxic activity was obtained at 48 h of treatment. 

### 2.3. Characterization of the Crude Extracts and Partitions from Miconia Chamissois

Analysis of the electrospray ionization fourier transform ion cyclotron resonance mass spectrometry (ESI (-) FT-ICR MS) profile of *M. chamissois* compounds mainly indicated the presence of phenolic compounds at a magnification of 100 to 700 *m/z* regions [34,35,36,37,38,39,40,41,42,43,44,45,46,47,48,49,50,51,52,53,54,55]. Among the compounds were Palmitic acid, linolenic acid, 5,7-Dihydroxy-2-(4-hydroxy-3,5-dimethoxyphenyl)-5,6-dihydro-4H-chromen-4-one, Methyl 4-[(3,4-dimethoxybenzoyl)oxy]-3-methoxybenzoate. The *m/z* values of the main molecules found in the *M. chamissois* extracts are shown in Table 6. 

In addition, the analysis of partition HP showed the presence of Ethyl gallate, Palmitic acid, and 6,7-Dimethoxy-1-(4-methoxy-phenyl)-isochroman-3-one. For CP, compounds such as Ethyl gallate, 2-(2-Ethoxy-2-oxoethyl)-2-hydroxysuccinic acid, 6,7-Dimethoxy-1-(4-methoxy-phenyl)-isochroman-3-one, Palmitic, and linoleic acid. Finally, AEP partition revealed Ethyl gallate, (2R,3R)-3,5-Dihydroxy-2-(3-hydroxy-4-methoxyphenyl)-7-methoxy-2,3-dihydro-4H-chromen-4-one,2-(2-Ethoxy-2-oxoethyl)-2-hydroxysuccinic acid, Kaempferol-7-neohesperidoside, 2,6-Bis-O-(3,4,5-trihydroxybenzoyl)-D-glucopyranose, 1,3,6-Trigalloyl glucose, 1,3,6-Trigalloyl glucose, quercitrin and Vitexin (Appendix A (Appendix A)).

## 3. Discussion

The Brazilian cerrado biome is a major source of plants, which has been explored for cancer treatment in traditional medicine [24,27,29,56]. Based on this, we performed a phytochemical screening. We provided data on the cytotoxic potential of leaf extracts from 13 plants of five families (Anacardiaceae, Annonaceae, Fabaceae, Melastomataceae, and Siparunaceae) on a panel of five glioma cell lines, indicating perspectives to developing new anticancer therapies.

The present study observed the poorest in vitro inhibitory activity for the *B. a ungulata, A. fraxinifolium*, *S. guianensis, B. variegata candida,* and *X. aromatica* crude leaf extracts toward the glioma cell lines. Therefore, although we have not observed cytotoxic effects for these crude extracts in gliomas, many may have action on other tumor types [57,58,59,60,61,62]. Recent studies published by our group revealed the importance of the diversity of secondary compounds in different crude extracts and their selective antitumor potential. Our group reported the antineoplastic effect of the Annonaceae family on cervical cancer [57]. The crude extract 7 and hexane partition from *Annona crassiflora* Mart. induced several biological responses, such as DNA damage and apoptosis, and also promoted a decrease in in vivo tumor perimeter in cervical cancer cell lines [63]. In addition, the hexane fraction of *X. aromatica* leaves constituted by phenolic acids, flavonoids, and alkaloids revealed antitumor potential on Ehrlich ascites carcinoma cell lines in vitro and in vivo [59]. Generally, the major components identified in Annonaceae are, typically, acetogenins [64], which exhibited in vitro anticancer activities [65,66], which could explain the anticancer activity observed for *A. crassiflora*, and other Annonaceae like *X. aromatica*. We also reported the cytotoxic potential of *Tapirira guianensis* Aubl extracts on a panel of head and neck squamous cell carcinoma cell lines [60]. In addition, we have previously demonstrated potential antimetastatic activity for *B. ungulata* extract fractions [58] and have described the mechanism of action of *B. variegata candida* fraction on cell viability and inhibitory activity against metalloproteinases in human cervical carcinoma (HeLa) and human peripheral blood mononuclear cells [61]. The present study emphasizes the importance of screening studies and provides a basis for further investigation on the antineoplastic activity of cerrado crude extracts antineoplastic activity on anticancer therapy.

Importantly, *M. cuspidata, M. albicans,* and *M. chamissois* revealed the greatest cytotoxic activity potential, and it is in agreement with the findings of the literature since their crude extracts belong to the Melastomataceae family, which is composed of important secondary compounds whose use in folk medicine has been focused on analgesics, anti-inflammatories, and even as antineoplastics [24,31,67]. These secondary compounds include simple phenolics, terpenoids, quinones, lignans, and their glycosides, as well as numerous tannins or polyphenols, some flavonoids, and anthocyanins [68]. The specie *M. albicans* from which crude extract 18 was obtained stands out for its anti-inflammatory, antimicrobial, and antiparasitic actions, as well as cytotoxic effects in some malignant neoplasms, as already evidenced [24,31]. Analgesic effects of *M. albicans* (Sw.) *Triana* of crude leaf extracts and their partitions (hexane, methylene chloride, and hydroalcoholic) present in leaves have also been described [67]. Despite the literature findings on the cytotoxic potential of some species of *Miconia,* to the best of our knowledge, no data related to *M. cuspidata* (extract 17) and *M. chamissois* (extract 19) has been previously published.

In this context, we selected crude extract number 19 from *M. chamissois* to continue the biological characterization for presenting the highest cytotoxic potential and for being one of the least cytotoxic to the normal NHA cells in this study. A better efficacy in the study of natural compounds and/or the effect of antineoplastic agents has been obtained from the partitioning and consequent fractionation of the crude extract to obtain pure components responsible for their bioactivity [69]. The cellular cytotoxicity studies of crude extract 19 and partitions (HDP (hydroalcoholic), HP (hexanic), CP (chloroform), and AEP (ethyl acetate) revealed that CP presented greater cytotoxic potential when compared to the crude extracts and the other partitions in glioma cells. A better characterization of the exposure time required for cytotoxic activity was also performed in the glioma cells through cell viability kinetics at 6, 12, 24, and 48 h. Crude extract 19, and its partitions, exhibited a dose and time-dependent cytotoxic response profile at 24 h of exposure, indicating acute cytotoxicity for these treatments. Studies with natural extracts from other plants on human glioma and colon tumor cells have also shown cytotoxic and antiproliferative effects at 24 h of treatment [70,71,72,73]. A comparison of IC_50_ values obtained through cell viability kinetics revealed that the CP partition was more cytotoxic than the crude extract and the HP and AEP partitions in most of the cell lines evaluated.

Understanding and exploiting the similarities and differences between adult and pediatric gliomas is an important strategy for developing new targeted therapies [2,3,4]. Here, we highlight the difference in cytotoxic effect between crude extracts in glioma cells, a heterogeneous response profile among adult and pediatric gliomas, and primary culture to crude extract and partitions exposition. It is known that pediatric GBM has a high molecular heterogeneity compared to adult GBM and within its group. All four high-grade lines have highly complex genomic profiles and harbored amplifications and deletions at several known cancer genes dysregulated in pediatric glioma. Thus, these factors may influence the aggressiveness of these tumors and the response to therapies [63,74,75,76]. Our results revealed GAMG more sensitive cells to treatments than other lines. According to the cosmic database, more than 280 mutations have already been found on the GAMG cell, revealing that different pathways may contribute to its sensitivity. However, our study constitutes an initial bioprospection cytotoxicity screening, and the tests carried out do not allow us to conclude the cell lines’ molecular response. This fact indicates a limitation in our study, indicating the need for molecular characterization and a proof concept that allows us to state the exact mechanisms underlying GAMG cells, making them more sensitive. Finally, we emphasize that 2D models are a starting point for studying new drugs and do not recapitulate the microenvironment and GBM’s complex landscape [7].

FT-ICR MS analysis revealed the phenolic compounds as the main molecules found in crude extract 19 and its partitions (HP, CP, and AEP), which were confirmed by similarity in a database [34,35,36,37,38,39,41,42,43,44,45,46,47,48,49,50,51,52,53,54,55]. Phenolic compounds are the biggest group of phytochemicals, and their antitumor potential and biological properties have been evaluated during the past years [77]. Phenolic compounds are widespread in cerrado plants, especially tannins, which are directly responsible for the therapeutic activity of plants in the cerrado biome [24,27,29,56]. Some of the compounds identified in the spectrum of partitions have been described concerning cytotoxicity activity and potential anticancer effects. Among them, oleic acid, the main ingredient of *Brucea javanica oil*), and widely known to have anticancer effects in many tumors [78]. Palmitic acid is well known for its antitumor effect in prostate cancer [79]. Cremastranone (5,7-dihydroxy-3-(3-hydroxy-4-methoxybenzyl)-6-methoxychroman-4-one), a homoisoflavanone isolated from *Cremastra appendiculata*, and it was reported by its anti-angiogenic and anti-proliferative activity in human umbilical vein endothelial cells and recently identified its property in suppression of growth of colorectal cancer cells through cell cycle arrest and induction of apoptosis [80]. Another compound that is highlighted is the Kaempferol-3-O-rhamnoside due to its properties in cancer and cell death [81]. Finally, although not mentioned here, many others are being actively characterized for their structure and antitumor potential.

We emphasize that most of a crude extract’s observed efficacy could result from a combined effect of more than one component. Therefore, the partition/fraction results may not always give a real efficacy of in vivo observation. Here, CP exhibited IC_50_ values significantly lower than the crude extract, suggesting a potential tumorigenic activity from CP and that a great part of the metabolites present in the crude extract responsible for the antitumor potential might be present in the chloroform partition.

Finally, recently our group harnessed the knowledge of this bioprospect study to assess the antitumor potential and biological characterization of CP on glioma cell lines [82]. We reported the effects of CP and its sub-fraction, McC1. The McC1 fraction significantly reduced glioma cell migration, invasion, and clonogenicity in vitro and reduced tumor growth and angiogenesis in vivo. Moreover, this component revealed synergistic effects with TMZ, indicating future perspectives on drug development [82].

## 4. Conclusions

Our study represents an important screening that systematically evaluated the anticancer activity and the selectivity index of 13 different phytochemical extracts from traditional Brazilian cerrado plant species against glioma cells. Furthermore, studies carried out with one of the most effective extracts (number 19 obtained from *M. chamissois* species) demonstrated that the chloroform partitions derived from this extract presented greater cytotoxicity in glioma cell lines, showing a heterogeneous and dose-time dependent response profile. Further studies are needed to confirm the potential and mechanisms of action of these extracts and partitions, yet, the results presented provided valuable information for the possible development of novel anticancer drugs based on these medicinal plants in the future.

## 5. Materials and Methods

### 5.1. Cell Culture

Five established central nervous system cell lines comprising two adults (GAMG and U251-MG) and two pediatric glioma cell lines (SF188 and RES259), and one normal human astrocyte (NHA) were obtained and cultivated, as described in Table 7. In addition, one primary cell culture (HCB151) derived from surgical glioblastoma biopsies at Barretos Cancer Hospital (São Paulo, Brazil) and previously established by our group was grown under the same conditions detailed in Table 7. The identification and confirmation were conducted on blood derived from the same patient. The Ethics Committee of Barretos Cancer Hospital approved this study [83]. Moreover, all glioma cells were analyzed for mycoplasma presence and authenticated by the Diagnostics Laboratory at Barretos Cancer Hospital through short tandem repeat (STR) DNA typing, as previously described [84].

### 5.2. Plant Extracts and Partitions

Leaves from 13 plant species of the Brazilian cerrado biome were collected in June 2015 (18°54′5066″ S and 48°13′5728″ W). The samples were identified, and voucher specimens of the plants were kept in the herbarium of the Federal University of Uberlândia (44998HUFU, 56558 HUFU, 59592HUFU), in the herbarium of the Federal University of Minas Gerais (143407BHCB, 143403BHCB, 43397BHCB, 143400BHCB, 143404BHCB, 161589BHCB, 161590BHCB, 161588BHCB) and in the herbarium of the Federal University of Grande Dourados (CG/MS) (under number 11486CG/MS) as presented in Table 8. The plant assays, including the collection of plant material, were conducted in accordance with local legislation. Detailed information about vernacular, botanical name, family, and registration code is identified in Table 8. The leaves were ground into powder, subjected to extraction, and then filtered. The extracts were freeze-dried to obtain the crude extracts. The partitioning from *M. chamissois* preparation was carried out using 4 different solvents with increased polarity: HDP-Hydroalcoholic, HP-Hexane, CP-Chloroform, and AEP-Ethyl acetate. The partitions were also submitted to freeze-drying. The crude extracts were diluted in dimethyl sulfoxide (DMSO) to obtain a final solution of 50 mg/mL and stored at −20 °C. The partitions were also dissolved in DMSO at a concentration of 25 mg/mL and stored at −20 °C. The intermediate dilutions of the crude extracts and partitions were prepared to obtain a concentration of 1% DMSO.

### 5.3. Cytotoxic Screening

#### 5.3.1. Cell Viability

The biological activity of crude extracts, partitions, and temozolomide (TMZ) (Sigma-T2577), were evaluated by Cell Titer 96 Aqueous cell proliferation assay (MTS assay, PROMEGA, Madison, WI, USA) as previously described [58,82]. Cells (5 × 10^3^) were plated into 96-well plates, exposed to the vehicle (1% DMSO) or increasing concentrations of the samples (crude extracts 1, 2, 3, and 7; 12 μg/mL to 300 μg/mL, crude extracts 8, 17, 18 and 19; 1.5 μg/mL to 300 μg/mL, crude extracts 10, 14I, 15I, 16I and 21I; 3 μg/mL to 300 μg/mL, partitions: 2.5 to 200 μg/mL, and TMZ, 1.21 to 48.54 µg/mL) diluted in DMEM medium (0.5% fetal bovine serum (FBS)) and incubated for 72 h. The samples’ absorbance was measured at 490 nm in an automatic microplate reader Varioskan (Thermo). The data were expressed as mean viable cells relative to DMSO alone (100% viability) ± SD. The experiments were performed in biological and experimental triplicates, and cell viability analyzes were calculated using the GraphPad PRISM version 5 program.

#### 5.3.2. Kinetics Assay

For the kinetics assay, glioma cells (5 × 10^3^) cell were plated into 96-well plates and exposed to the vehicle (1% DMSO) or increasing concentrations of extract 19 partitions (concentrations from 1.5 µg/mL to 100 µg/mL), diluted in DMEM medium (0.5% fetal bovine serum (FBS)) and incubated for different time points (6, 12, 24, 48 h). The sample’s absorbance was measured at 490 nm in an automatic microplate reader Varioskan (Thermo). The data were expressed as mean viable cells relative to DMSO alone (100% viability) ± SD. The experiments were performed in biological and experimental triplicates, and cell viability analyzes were calculated using the GraphPad PRISM version 5 program.

#### 5.3.3. Inhibitory Concentration of 50% of Cells (IC_50_)

IC_50_ determination (inhibitory concentration of 50% of cells) was performed by non-linear regression analysis using the GraphPad PRISM software version 5 from the viability results obtained in item 5.3.1. The NCI (American National Cancer Institute) determines that promising crude extracts for subsequent purification studies and cytotoxicity assays have IC_50_ values < 30 μg/mL, and for the cytotoxicity studies of this work, this same previously established criterion was adopted [33].

### 5.4. Selectivity Index

Crude extracts and partitions, as well as TMZ, were also evaluated by the selectivity index (SI), which allows identifying the selectivity of the tested compounds and their potential use in clinical studies. In this study, SI was obtained from the following formula: IC_50_ of non-tumor cell line (NHA)/IC_50_ of tumor cell lines. For this analysis, an SI value greater than or equal to 2.0 was adopted as significant, as previously described by the NCI (National Cancer Institute) [33].

### 5.5. Analysis of Secondary Compounds Present in the Crude Extract and Partitions in Miconia Chamissois by FT-ICR MS

The crude extract and partitions were analyzed using the negative ion-mode Electrospray Ionization Fourier Transform Ion Cyclotron Resonance Mass Spectrometer (ESI (−) FT-ICR MS, model 9.4 T Solarix, Bruker Daltonics Bremen). All mass spectra were externally calibrated using NaTFA (*m/z* from 200 to 2000). The source parameters of the ESI (−) source were: nebulizer gas pressure of 0.5–1.0 bar, capillary voltage of 3–3.5 Kv, and capillary transfer temperature of 250 °C. The resolution power used was m/Δm50% ≅ 200,000 (where Δm50% is the maximum peak width at peak height *m/z* ≅ 400) and mass accuracy <8 ppm. The degree of unsaturation for each molecule can be deduced directly from its DBE value according to the equation DBE = c − h/2 + *n*/2 + 1, where c, h, and *n* are the numbers of carbon atoms, hydrogens, and nitrogen in the molecular formula, respectively. The FT-ICR mass spectrum was acquired and processed using Compass Data Analysis software. The elemental compositions of the present compounds were determined by measuring the *m/z* ratio values. The (ESI (−) FT-ICR MS and NMR spectrum are presented in the Appendix A.

### 5.6. Statistical Analysis

The data were expressed as the mean ± standard deviation (SD) of at least three independent experiments. The Student’s *t*-test was applied to compare two different conditions, whereas a two-way analysis of variance (ANOVA) was used for assessing differences between more groups, and a *p* value < 0.05 was adopted as a statistically significant difference. All the analyses were performed using the aforementioned GraphPad PRISM version 7 (GraphPad Software, La Jolla, CA, USA).

## Figures and Tables

**Figure 1 molecules-28-01394-f001:**
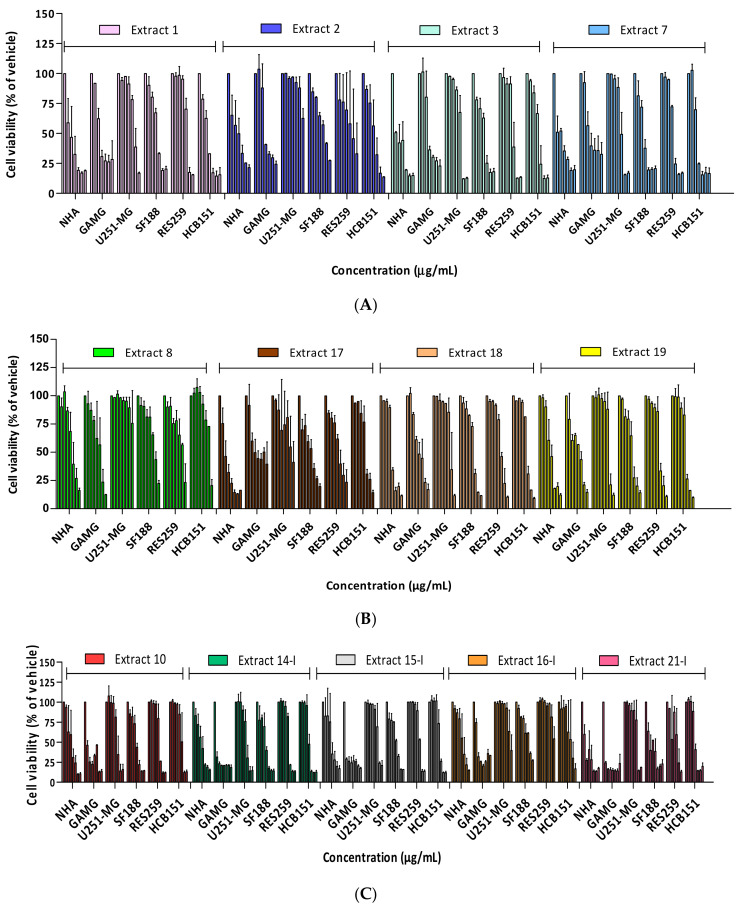
Time and concentration cytotoxic effect of the crude extract’s exposure on the human cancer cell line. Cells (5 × 10^3^) were plated into 96-well plates, exposed to the vehicle (1% DMSO) or increasing concentrations of the samples (**A**) crude extracts (1, 2, 3 and 7) 12 μg/mL to 300 μg/mL, (**B**) crude extracts (8, 17, 18 and 19) 1.5 μg/mL to 300 μg/mL and (**C**) crude extracts (10, 14I, 15I, 16I and 21I) 3 μg/mL to 300 μg/mL. Cell viability was measured with an MTS assay after 72 h of extract treatment. The cell viability of the untreated cells was regarded as 100%. The results shown are the means ± S.D of three independent experiments.

**Table 1 molecules-28-01394-t001:** Mean and standard deviation of IC_50_ (µg/mL) values for crude leaf extracts from plants of the Brazilian cerrado biome and the chemotherapy temozolomide, evaluated at 72 h.

IC_50_ (µg/mL) Mean ± Standard Deviation (SD)						
Crude Extract/Chemotherapy	NHA	GAMG	U251-MG	SF188	RES259	HCB151
1	36.38 ± 11.3	<1.5	146.1 ± 35.8	72.72 ± 8.3	89.92 ± 71.5	25.93 ± 12.9
2	13.4 ± 2.3	<1.5	>300	118.5 ± 1.7	124.7 ± 188.1	72.85 ± 26.9
3	>200	<1.5	120.6 ± 10.9	57.45 ± 13.7	70.93 ± 40	54.78 ± 22
7	41.30 ± 18.21	<1.5	117.2 ± 53.5	43.75 ± 1.4	50.40 ± 39	28.69 ± 13.4
8	36.19 ± 20.1	6.66 ± 4.3	176.5 ± 74.88	66.20 ± 5.3	86.80 ± 43.5	125.7 ± 10.9
10	21.46 ± 1.9	18.83 ± 4.67	70.52 ± 61	18.45 ± 16.9	44.15 ± 43.9	145.7 ± 19.7
14-I	22.84 ± 14.05	<1.5	77.68 ± 20.1	26.55 ± 12.4	74.76 ± 1.5	55.30 ± 0
15-I	35.68 ± 37.1	<1.5	136.4 ± 33.6	34.31 ± 8	106.3 ± 5.2	81.95 ± 0
16-I	62.67 ± 50.2	<1.5	296.9 ± 23.2	100.5 ± 16.6	>200	208.1 ± 30.1
17	2.9 ± 1.4	25.7 ± 12.9	28.9 ± 5.5	13.6 ± 5.7	29.3 ± 17.4	30.2 ± 10.6
18	12.05 ± 1.1	19.94 ± 0.4	70.21 ± 7.4	23.08 ± 3.6	38.07 ± 5.7	30.05 ± 3.9
19	12.11 ± 5.3	8.37 ± 11.8	89.85 ± 5	19.28 ± 8.3	33.04 ± 12.0	27.43 ± 6.2
21-I	3.73 ± 3.7	<1.5	132.5 ± 32.5	6.56 ± 0.6	37.04 ± 8.4	47.52 ± 0
TMZ	1.28 ± 26.26	7.06 ± 31.84	39.78 ± 23.67	30.74 ± 27.91	16.93 ± 27.55	38.22 ± 26.02

TMZ (temozolomide).

**Table 2 molecules-28-01394-t002:** Selectivity index for crude leaf extracts from plants of Brazilian cerrado biome and chemotherapy temozolomide.

Crude Extract/Chemotherapy	Glioma Cell Lines				
	GAMG	U251-MG	SF188	RES259	HCB151
1	*UD*	0.2	0.5	0.4	1.4
2	*UD*	*UD*	0.1	0.1	0.2
3	*UD*	*ID*	*UD*	*UD*	*UD*
7	*UD*	0.4	0.9	0.8	1.4
8	5.4	0.2	0.5	0.4	0.3
10	1.1	0.3	1.2	0.5	0.1
14-I	*UD*	0.3	0.9	0.3	0.4
15-I	*UD*	0.3	1.0	0.3	0.4
16-I	*UD*	0.2	0.6	*UD*	0.3
17	0.1	0.1	0.2	0.1	0.1
18	0.6	0.2	0.5	0.3	0.4
19	1.4	0.1	0.6	0.4	0.4
21-I	*UD*	0.03	0.6	0.1	0.1
TMZ	0.2	0.03	0.0	0.1	0.03

*UD—*undetermined values. TMZ (temozolomide).

**Table 3 molecules-28-01394-t003:** IC_50_ values for crude extract and partitions from *Miconia chamissois* and chemotherapy, temozolomide, evaluated at 72 h.

IC_50_ (µg/mL) Mean ± Standard Deviation (SD)						
Crude Extract/Partitions/Chemotherapy	NHA	GAMG	U251-MG	SF188	RES259	HCB151
CE	12.11 ± 5.35	8.37 ± 11.81	89.85 ± 5.09	19.28 ± 8.28	33.04 ± 12.08	27.43 ± 6.20
HDP	>200	>200	>200	>200	>200	>200
HP	9.17 ± 0.99	37.39 ± 5.22	34.66 ± 5.45	5.35 ± 2.27	18.22 ± 4.58	21.76 ± 6.74
CP	14.06 ± 2.41	34.31 ± 5.50	21.54 ± 1.80	7.28 ± 3.40	20.83 ± 4.91	15.00 ± 1.00
AEP	31.22 ± 10.40	2.57 ± 3.34	102.4 ± 2.61	41.69 ± 6.06	27.40 ± 4.25	34.26 ± 4.20
TMZ	1.28 ± 26.26	7.06 ± 31.84	39.78 ± 23.67	30.74 ± 27.91	16.93 ± 27.55	38.22 ± 26.02

CE (crude extract); HDP (hydroalcoholic partition); HP (hexane partition); CP (chloroform partition); AEP (ethyl acetate partition); temozolomide (TMZ).

**Table 4 molecules-28-01394-t004:** Selectivity index for crude extracts and partitions from *Miconia chamissois* and chemotherapy, temozolomide.

Crude Extract/Partitions/Chemotherapy	Cell Lines				
	GAMG	U251-MG	SF188	RES259	HCB151
CE	1.4	0.1	0.6	0.4	0.4
HDP	*UD*	*UD*	*UD*	*UD*	*UD*
HP	0.2	0.3	1.7	0.5	0.4
CP	0.4	0.7	1.9	0.7	0.9
AEP	12.1	0.3	0.7	1.1	0.9
TMZ	0.2	0.03	0.04	0.1	0.03

*UD*—undetermined values. CE (crude extract); HDP (hydroalcoholic partition); HP (hexane partition); CP (chloroform partition); AEP (ethyl acetate partition); temozolomide (TMZ).

**Table 5 molecules-28-01394-t005:** IC_50_ values for crude leaf extract and partitions from *Miconia chamissois*.

IC_50_ (µg/mL) Mean ± Standard Deviation (SD)					
Cell Lines	Crude Extract/Partition	6 h	12 h	24 h	48 h
NHA	CE	>300	>300	>300	64.94 ± 23.29
	HP	>200	77.66 ± 24.02	37.64 ± 32.21	30.55 ± 37.02
	CP	>200	61.39 ± 21.98	35.13 ± 26.58	20.93 ± 32.62
GAMG	CE	>300	>300	89.29 ± 20.33	35.98 ± 26.61
	HP	>200	49.20 ± 21.85	55.95 ± 26.24	34.78 ± 32.96
	CP	51.52 ± 18.43	32.17 ± 24.64	40.11 ± 26.13	39.14 ± 23.52
U251-MG	CE	>300	>300	>300	>300
	HP	>200	>200	>200	>200
	CP	>200	>200	>200	69.03 ± 30.89
SF188	CE	>300	91.88 ± 17.86	56.87 ± 24.38	15.42 ± 32.78
	HP	>200	51.88 ± 24.84	26.54 ± 31.35	16.46 ± 34.98
	CP	>200	41.58 ± 22.61	22.35 ± 30.20	6.34 ± 32.81
RES259	CE	>300	>300	>300	20.45 ± 24.31
	HP	>200	93.91 ± 19.89	66.21 ± 30.07	22.89 ± 32.60
	CP	>200	>200	64.72 ± 25.95	9.16 ± 35.78

CE (crude extract), HP (hexane partition) and CP (chloroform partition).

**Table 6 molecules-28-01394-t006:** Proposed structures by ESI (-) FT-ICR MS for the main molecules of *Miconia chamissois* crude extract.

*m/z* Measured	*m/z* Theoretical	Error (ppm)	DBE	[M-H]	Proposed Compound	Reference
191.05620	191.05602	−0.45	2	[C_7_H_11_O_6_]	chlorogenic acid	[34]
197.04549	197.16511	0.27	5	[C_9_H_9_O_5_]	syringic acid	[35]
219.05098	219.16907	0.23	3	[C_8_H_11_O_7_]	2-hydroxypropane-1,2,3-tricarboxylic acid, dimethyl ester	[36]
229.07151	229.07130	1.11	4	[C_10_H_13_O_6_]	D-(-) Erythrose	[37]
255.23294	255.23295	0.06	1	[C_16_H_31_O_2_]	palmitic acid	[38]
265.10783	266.10789	1.19	6	[C_14_H_17_O_5_^]^	goniothalesdiol	[39]
277.21771	277.25421	−1.46	4	[C_19_H_29_O_2_]	linolenic acid	[40]
281.24860	281.45410	0.01	2	[C_18_H_33_O_2_]	oleic acid	[41]
299.09247	299.29864	0.11	10	[C_17_H_15_O_5_]	5-hydroxy-6,7-dimethoxyflavanone	[42]
309.09756	311.37850	1.35	7	[C_15_H_17_O_7_]	picrotin	[43]
313.10808	313.32526	0.20	10	[C_18_H_17_O_5_]	agrimonolide	[44]
331.08231	331.29745	0.06	10	[C_17_H_15_O_7_]	dihydroquercetin-7,4’-dimethylether	[45]
345.09851	345.32300	−1.55	10	[C_18_H_17_O_7_]	cremastranone	[46]
351.12974	351.32698	−0.19	3	[C_14_H_23_O_10_]	methyl 4-(6.7-dideoxy-galacto-hept-6-enopyranosyl)-galactopyranoside	[47]
363.10909	363.10800	−1.52	9	[C_18_H_19_O_8_]	4,4’-Dihydroxy-3,3’-dimethoxy-2,2’-dimethyl-5,6,5’,6’-bimethylenedioxybiphenyl	[48]
379.13921	375.13410	1.67	8	[C_19_H_23_O_8_]	gibberellic Acid	[49]
431.09823	431.37037	0.33	12	[C_21_H_19_O_10_]	kaempferol-o-rhamnoside	[50]
445.13427	445.13400	1.97	7	[C_19_H_25_O_12_]	canthoside A	[51]
447.09333	447.36978	−0.09	12	[C_21_H_19_O_11_]	6-hydroxyapigenin-7-o-β-d-glucopyranoside	[52]
455.35390	455.35100	−1.82	7	[C_30_H_47_O_3_]	3-Oxotirucalla-7,24-dien-21-oic acid	[53]
483.07802	483.35726	0.02	11	[C_20_H_19_O_14_]	3, 6-di-o-galloyl-d-glucose	[54]
593.15123	593.51122	−0.06	13	[C_27_H_29_O_15_]	rutinosylkaempferol	[55]
635.08917	635.46179	−0.28	16	[C_27_H_23_O_18_]	1,2,3-tri-o-galloyl-β-d-glucose	[54]

DBE: Double bond equivalent; *m/z*: mass-to-charge ratio.

**Table 7 molecules-28-01394-t007:** Culture conditions and origin of the cell lines used.

Cell Lines	Type	Subtype	Grade (OMS)	Description	Sex	Age	Mainly Mutations	Origin	Culture Conditions
GAMG	Established	GBM	IV	Adult	Female	42 years	*TP53 and TERT*	DSMZ	DMEM + 10% FBS + 1% *P*/*S*
U251-MG	Established	GBM	IV	Adult	Male	75 years	*CDKN2A/B*, *EGFR, TP53*, *PTEN and TERT*	Kindly provided by Dr. Joseph Costello	DMEM + 10% FBS + 1% *P*/*S*
SF188	Established	GBM	IV	Paediatric	Male	8 years	*TP53 and NF1*	Kindly provided by Dr. Chris Jones	DMEM + 10% FBS + 1% *P*/*S*
RES259	Established	DA	II	Paediatric	Female	4 years	*TERT*	Kindly provided by Dr. Chris Jones	DMEM + 10% FBS + 1% *P*/*S*
HCB151	Primary	GBM	IV	Adult	Male	59 years	*CDKN2A and PTEN*	Barretos Cancer Hospital	DMEM + 10% FBS + 1% *P*/*S*
NHA	Established	-	-	NA	NA	NA	-	ECACC	DMEM + 10% FBS + 1% *P*/*S*

GBM: Glioblastoma, DA: Diffuse astrocytoma; NA = Not available; NHA: Normal Human Astrocyte, DSMZ—German Collection of Microorganisms and Cell Cultures. ECACC (European Collection of Authenticated Cultures). FBS: Fetal Bovine Serum; *P*/*S* Penicillin/Streptomycin solution.

**Table 8 molecules-28-01394-t008:** Classification of natural extracts from the Brazilian cerrado biome.

Crude Extract Identification	Vernacular Name	Botanical Name	Family	Registration Code
1	Pau Pombo	*Tapirira guianensis*	*Fabaceae*	143407BHCB
2	Gonçalo-Alves	*Astronium fraxinifolium*	*Anacardiaceae*	143403BHCB
3	Pimenta-de-Macaco	*Xylopia aromatica*	*Annonaceae*	43397BHCB
7	Araticum	*Annona crassiflora*	*Annonaceae*	143400BHCB
8	Negramina	*Siparuna guianensis*	*Siparunaceae*	143404BHCB
10	Marcela	*Achyrocline alata*	*Asteraceae*	11486CG/MS
14-I	Pata-de-vaca	*Bauhinia variegata*	*Fabaceae*	161589BHCB
15-I	Pata-de-vaca branca	*Bauhinia variegata candida*	*Fabaceae*	161590BHCB
16-I	Pata-de-viado	*Bauhinia ungulata*	*Fabaceae*	161588BHCB
17	Pixirica-da-mata	*Miconia cuspidata*	*Melastomataceae*	44998HUFU
18	Canela de velho	*Miconia albicans*	*Melastomataceae*	56558 HUFU
19	Pixirica-açu	*Miconia chamissois*	*Melastomataceae*	59592HUFU
21-I	Barbatimão	*Stryphnodendron adstringens*	*Fabaceae*	169871BHCB

HUFU—herbarium of the Federal University of Uberlândia; BHCB—herbarium of the Federal University of Minas Gerais; CG/MS—herbarium of the Federal University of Grande Dourados.

## Data Availability

Not applicable.

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
