# Peer review of "Bio-Prospecting of Crude Leaf Extracts from Thirteen Plants of Brazilian Cerrado Biome on Human Glioma Cell Lines"

_molecules, 2023, doi:10.3390/molecules28031394_

Round 1

Reviewer 1 Report

In this manuscript, the anticancer activity and selectivity index of the different phytochemical extracts from 13 traditional Brazilian cerrado plant species of 5 families (Anacardiaceae, Annonaceae, Fabaceae, Melastomataceae Siparunaceae) against glioma cells were systematically evaluated. Moreover, the mass spectrometry (ESI FT-ICR MS) was performed to identified the secondary metabolites classes presented in the crude extracts and partitions. Here are some suggestions for further revision.

1.      Please provide more detailed background introduction and study significance in the introduction section.

2.      Please provide some figures in the Results section.

3.      Was the cytotoxicity of the extraction solvent considered?

4.      Whether there are differences in the types of secondary compounds in different solvent extracts?

5.      Whether the PCA, PLS-DA, OPLS-DA and other methods can be used to explore the reasons for the differences in cytotoxicity from different extraction solvent?

6.      Please provide more detailed FT-ICR MS identification information in the supplementary materials.

7.      It is suggested to provide a separated Conclusions section.

8.      There are some mistakes in format, please check and revised carefully. For example, “m/z”, “table 8”.

Author Response

Dear Reviewer,

We appreciate the opportunity to clarify our manuscript message and the reviewer’s insightful comments. The manuscript has been reviewed following editorial board comments/recommendations. All corrections in the manuscript are highlighted in yellow. Please see the attachment below for a detailed point-by-point response to the reviewer’s comments.

Sincerely,

Rui M Reis

Reviewer 2 Report

The authors present a Screening of potential cytotoxic activities of 13 traditional plant species from the Brazilian cerrado bioma against 5 glioma cells. The study on 5 cell lines stands out, as well as the inclusion of normal astrocytic cells to establish the selectivity index.

However, the article should be carefully reviewed in the following aspects:

The cells must be described in their particular characteristics, which could help to improve the discussion of the results since a generalization of the activity on glioblastoma is made. Still, very different results are observed between the lines used.

The abstract should be reformulated in the following comments: line 25: “Treatment remains essentially palliative”; line 38: “The partitions also showed a mean IC50 close to chemotherapy, temozolomide”; line 42: “…promising component”

The described method is concise but not precise enough and the conclusion should bring the focus back to the aim.

 The introduction is very poor, it should include, among others, the anticancer mechanism of temozolomide, data from Plant to understand selection criteria

 In the results section:

Several tables are shown that indicate the IC50 values ​​for each line, however at the end, an average of these values ​​is made, this average does not present an SD, on the other hand, it suggests a generalization of the result that is not adequate since they are observed different results between cells lines, which suggests an effect dependent on the particular characteristics of each one.

An analysis of the selectivity index is carried out; however, this result does not seem to influence the inclusion criteria for the fractionation of the crude extract #19 that was chosen as promising.

Lines 95-96: Indicate here the corresponding numbers of the extracts

The comparison between the IC50 values of the extracts and the TMZ control for healthy cells is not appropriate, this analysis should be done about the selectivity index.

It is not understood why some data appear as <1.5, could not this value be recalculated, which could represent a better cytotoxic activity and even a better selectivity index for those extracts?

Table #3 indicate the exposure time.

AEP was not included as part of the kinetics, why?

In Table 5: Why was the kinetics performed up to 48 h of treatment if the study uses a time of up to 72 h to determine cytotoxicity?

Table 6 shows a list of components that are not discriminated according to their origin: crude extract, and fractions.

In the discussion:

Too much theoretical data is included but no clear explanation is given regarding the results, why are GAMG cells more sensitive to treatments than other lines?

Which of the metabolites found in the active fractions could be involved in the cytotoxic activity?

Author Response

(The authors gave the same response as above.)

Round 2

Reviewer 1 Report

The comments have been well addressed, and the manuscript has been carefully revised. This manuscript may be accepted for publication after minor revisions.

1, Please carefully check the Latin name of the plants. For example, the abbreviation “M.” can be used for “Miconia” in the Latin names after its first used in the text.

2, The “m/z” should be in Italic font-style.

3, Please check the Table 6, is that necessary to cite a reference for each identified compound?

Author Response

Dear Review,

Please see the attachment our revised manuscript. We appreciate the reviewer’s insightful comments and the opportunity to clarify our manuscript message. 

Sincerely,

Rui Reis

Reviewer 2 Report

The paper was adjusted according to the recommendations. Thanks to the author for improving the information described here

Author Response

(The authors gave the same response as above.)
